# Authentication of Transylvanian Spirits Based on Isotope and Elemental Signatures in Conjunction with Statistical Methods

**DOI:** 10.3390/foods10123000

**Published:** 2021-12-04

**Authors:** Dana Alina Magdas, Gabriela Cristea, Adrian Pîrnau, Ioana Feher, Ariana Raluca Hategan, Adriana Dehelean

**Affiliations:** National Institute for Research and Development of Isotopic and Molecular Technologies, P.O. Box 700, 400293 Cluj-Napoca, Romania; gabriela.cristea@itim-cj.ro (G.C.); adrian.pirnau@itim-cj.ro (A.P.); ioana.feher@itim-cj.ro (I.F.); ariana.hategan@itim-cj.ro (A.R.H.); adriana.dehelean@itim-cj.ro (A.D.)

**Keywords:** fruit distillates, stable isotopes, elemental content, LDA, discrimination

## Abstract

The potential association between stable isotope ratios of light elements and mineral content, in conjunction with unsupervised and supervised statistical methods, for differentiation of spirits, with respect to some previously defined criteria, is reviewed in this work. Thus, based on linear discriminant analysis (LDA), it was possible to differentiate the geographical origin of distillates in a percentage of 96.2% for the initial validation, and the cross-validation step of the method returned 84.6% of correctly classified samples. An excellent separation was also obtained for the differentiation of spirits producers, 100% in initial classification, and 95.7% in cross-validation, respectively. For the varietal recognition, the best differentiation was achieved for apricot and pear distillates, a 100% discrimination being obtained in both classifications (initial and cross-validation). Good classification percentages were also obtained for plum and apple distillates, where models with 88.2% and 82.4% in initial and cross-validation, respectively, were achieved for plum differentiation. A similar value in the cross-validation procedure was reached for the apple spirits. The lowest classification percent was obtained for quince distillates (76.5% in initial classification followed by 70.4% in cross-validation). Our results have high practical importance, especially for trademark recognition, taking into account that fruit distillates are high-value commodities; therefore, the temptation of “fraud”, i.e., by passing regular distillates as branded ones, could occur.

## 1. Introduction

In East Europe, there are some countries well known for their tradition in the production of fruit spirits, e.g., Hungary (pálinka), Bulgaria (slivovarakya), Poland (śliwowica Łącka), Slovakia (bošácka slivovica), and the Czech Republic (slivovice) [1]. Moreover, Romania follows an “old tradition” regarding the production of these kinds of alcoholic drinks. Nearly all Romanian traditional strong drinks are made from fruits, which technically make them “brandy”. The term “brandy” refers to any alcohol made by fermentation and distillation of fruits. The most appreciated fruits that are used in the production of spirits in Romania, particularly in the Transylvania region, are plums. Fermenting and twice distilling plums results in the nationally famous drink “tuică”, having 24–65% alcohol volume. It is estimated that over 75% of all plums harvested in Romania are used to make “ţuica”. “Pălincă” is another traditional Romanian alcoholic beverage obtained exclusively by alcoholic fermentation and distillation of a fleshy fruit or a mixture of fruits, except plums.

The compositions of fruit distillates are complex, no matter what fruit types are utilized, because of the various natural factors that influence their specific fingerprints (i.e., fruit variety, geographical origin), and due to the specific elaboration technologies. Thus, besides the natural conditions that are specific to the used raw materials, the distillation process, followed by an aging process, brings a special personality to each produced distillate. Therefore, the distillate elaboration is crucial for the final taste and aroma of the fruit spirits.

Isotope-ratio mass spectrometry (IRMS) and SNIF-NMR are popular techniques used to verify the authenticity of grains and fruit spirits [2]. The isotope fingerprints of ^13^C, ^2^H, and ^18^O measured through IRMS, along with (D/H)_I_ and (D/H)_II_ ratios determined through ^2^H-NMR, are good markers for authentication of grain and fruit spirits [3,4,5]. Because numerous types of raw materials are used in distillate production, the authentication of spirits is a broad domain, requiring extensive studies to cover the large variability given by the fruit types. Another main limitation involves the different technologies used by distinct producers for elaboration of fruit distillates, which contribute to a specific technological fingerprint to the final signature of the spirits [6]. Finally, the purchase of monovarietal distillate samples from a specific area is difficult to perform, especially when the samples are supposed to come from the same producer. This is because, in many cases, manufactured distillates are elaborated from a mixture of fruits. Therefore, as mentioned in the literature, most of the scientific studies on spirits are conducted based on a limited number of samples [7].

To our knowledge, for Romanian fruit distillates, most of the published studies were related to the physicochemical proprieties, the antioxidant content, and minor and major volatile compounds [1,8], with no isotope data being reported until now. Apart from these studies, which were based on differentiation markers, other emerging differentiation approaches were reported. In this regard, an example would be the discrimination of Romanian distillates based on Raman spectroscopy and machine learning algorithms that were recently reported on by our group [6].

To find a suitable approach that could differentiate fruit distillates according to the geographical origin and the fruit variety and to test the possibility of identifying the trademark fingerprint, the classification potential of the isotope and elemental composition of spirits was tested in this study. All analyzed samples used for this study originated from Transylvania, Romania.

## 2. Experimental

### 2.1. Sample Description

A total of 26 distillate samples (500 mL each) were obtained from distinct raw materials, as follows: plums (*Prunus domestica*) (5), apricots (*Prunus armeniaca*) (3), apples (*Malus domestica*) (3), grapes (*Vitis vinifera*) (3), pears (*Pyrus communis* L.) (3), quinces (*Cydonia oblonga*) (3), bitter cherries (*Prunus emarginata*) (2), blueberries blackcurrant (*Ribes nigrum*) (1), sour cherries (*Prunus cerasus*), beer (1) and fruits mixture (1). They were collected and analyzed in the frame of this study. All investigated samples were purchased from two distinct processors and one manufacture, located in Transylvania, Romania. A detailed description of the sample distributions is provided in Table A1.

### 2.2. Isotope Measurements

All isotope measurements were performed on the ethanol recovered after the distillation of the fruit spirits. The extraction of the ethanol was performed with a distillation column with a rotating Teflon band (Micro Spinning Band Column–NORMAG), designed for optimal conditions of high-precision distillation that does not lead to isotopic fractionations. After the sample distillation, the alcoholic strength was determined, using an electronic densitometer (Rudolph Research DDM 2910).

#### 2.2.1. IRMS—Isotope-Ratio Mass Spectrometry

##### ^13^C Measurements

For the δ^13^C measurements, 8 µL of ethanol of each sample was combusted at 550 °C, using a Nabertherm oven (Germany) for 3 h to obtain CO_2_. After combustion, the resulting CO_2_ was purified from the other combustion gasses by cryogenic separation. Then, carbon isotope ratios of CO_2_ were determined using an isotope ratios mass spectrometer (Delta V Advantage, Thermo Scientific, Waltham, MA, USA) in line with a dual inlet system, by simultaneous recordings of masses 44 (^12^C^16^O_2_), 45 (^13^C^12^C^16^O, ^12^C_2_^17^O), and 46 (^12^C^16^O^18^O). All ethanol samples were measured in duplicate. Every day, one working standard was measured before the analysis of the fruit distillates, which was previously calibrated against NBS-22 oil (IAEA—International Atomic Energy Agency, Vienna, Austria), having a certified value of δ^13^C_VPDB_ = −30.03 ‰. The limit of uncertainty was ±0.3 ‰ for δ^13^C determinations.

##### ^2^H and ^18^O Measurements

To measure ^2^H/^1^H and ^18^O/^16^O isotope ratios, the first performed step consisted of the obtainment of ethanol from fruit spirit distillation. The extraction of ethanol was performed using a Cadiot spinning band column. The remaining water was trapped by preserving the distillate for 48 h on a molecular sieve, 3 Å (beads, 8–12 mesh, Sigma-Aldrich, St. Louis, MO, USA), as previously described and discussed in the literature [9,10]. In the next step, a high-temperature pyrolysis system of the elemental analyzer (Flash EA 1112 HT) coupled to an isotope ratio mass spectrometer (Delta V Advantage, Thermo Scientific) was used. To inlet the samples in the liquid injector (AI 1310 Thermo Scientific), 2 mL vials were filled with fruit distillates. The reactor temperature was set at 1400 °C. The carrier gas was He (99.9999% purity) and its pressure in the reactor was 1.4 bar. The working standards were H_2_ and CO, which were calibrated versus Vienna Standard Ocean Mean Water (VSOMW) international standards (δ^2^H_VSMOW_ = 0‰, δ^18^O_VSMOW_ = 0‰) by analyzing the GISP (δ^2^H_VSMOW_ = −189.5‰, δ^18^O_VSMOW_ = −24.76 ‰) and SLAP2 (δ^2^H_VSMOW_ = −427.5‰, δ^18^O_VSMOW_ = −55.5 ‰) (IAEA) international reference materials. The limit of uncertainty was ±1.0 ‰ for δ^2^H, and ±0.3 for δ^18^O. All samples were measured in triplicates.

The isotope compositions are denoted in delta values versus international standards (V-PDB—Vienna Pee Dee Belemnite for carbon; VSOMW—Vienna Standard Ocean Mean Water for hydrogen and oxygen), according to the equation [11]:(1)$$\delta^{i}X=\frac{R_{sample}}{R_{standard}}-1$$
where *i* is the mass number of the heavier isotope of the element *X* (^13^C, ^2^H, ^18^O), *R_sample_* is the isotope number ratio of a sample (^13^C/^12^C; ^2^H/^1^H; ^18^O/^16^O), and *R_standard_* is that of the international standard. The delta values are multiplied by 1000 and are expressed in units “per mil” (‰).

#### 2.2.2. NMR Measurements

##### Samples Preparation for NMR Analysis

For deuterium NMR measurements, a sample contains: 1.3 mL tetramethylurea (TMU) provided by IRMM Geel Belgium, which is used as a standard with a known and verified isotopic ratio (D/H); 0.2 mL hexafluorobenzene (C_6_F_6_), used for the stability of the magnetic field NMR spectrometer and 3 mL ethanol extracted from fruit distillates. The three components of the sample were accurately weighed, put together in a high-resolution 10 mm NMR tube, and the resulted mixture was homogenized by shaking.

The NMR measurements were performed with the BRUKER Avance III 500 UltraShield NMR spectrometer equipped with a 10 mm deuterium probe head (fluorine lock) ^2^H -^1^H -^19^F type SEX 500 MHz S2, at a deuterium resonance frequency of 76.7 MHz, corresponding to the magnetic field of the superconducting magnet of B_0_ = 11.7 T. All NMR measurements were performed using the following acquisition parameters: spectral width 12 ppm, data points 16 K, 23.5 μs for 90° pulse width, 256 number of scans, 8.89 s acquisition time, 11 s relaxation delay, and the temperature was kipped to 302 K. The calculation of D/H of ethanol was performed using the method described in the Council Regulation EEC 2676/90.

The D/H ratios of ethanol from the fruit distillate samples were determined according to the official analytical method for wine analysis by quantitative NMR spectroscopy. A deuterium natural abundance quantitative NMR method (SNIF-NMR: site-specific natural isotope fractionation) was developed as an efficient and capable means of characterizing the chemical origins (natural or synthetic) of the organic molecules and of distinguishing their biological and geographical origins. The method was developed as described by the Council Regulation EEC 2676/90. This method is based on the measurement of the (D/H) ratios at the methyl (D/H)_I_ and methylene (D/H)_II_ sites of the ethanol molecule. (D/H)_I_ mainly characterizes the vegetable species, which synthesizes the sugar and, to a lesser extent, the geographical location of the place of harvest (type of water used during photosynthesis); (D/H)_II_ represents the climatology of the place of production.

### 2.3. ICP-MS Determinations

ICP-MS was used for multi-elemental analysis of distilled beverages using an ELAN DRC (e) (Perkin Elmer) mass spectrometer with a Meinhard nebulizer and silica cyclonic spray chamber. The optimized parameters for the ICP-MS measurements were the radiofrequency generator power output: 1000 W; argon flows: plasma, 17 L/min; nebulizer: 0.93 L/min; auxiliary gas flow: 1.4 L/min; CeO/Ce = 0.020; Ba++/Ba = 0.023. The chosen conditions were a compromise between the highest 103Rh ion signal and the lowest percentage of doubly charged ions, obtained by the intensity’s ratio Ba++/Ba (always ≤3%) and of oxide ions, obtained by the intensity’s ratio CeO/Ce, always ≤3%; precision better than 2% and background <30 cps. The operating conditions were optimized daily using a solution containing 10 μg/L of Mg, Ba, Ce, Cu, Cd, Rh, In, and Pb, and monitoring the intensities at mass 69, 156, and 220, corresponding to species 138Ba^2+^, 140Ce16O^+^, and background, respectively.

To avoid the high alcohol content of distillate samples from precluding the plasma ignition, each sample was diluted with 2% nitric acid solution (*v*/*v*) prepared in ultrapure water (Simplicity^®^ UV System, Millipore) in order to obtain an ethanol solution of 2% (*v*/*v*). Before the accurate quantification, the concentrations of each element were analyzed by a Total Quant approach, using a solution of 10 μg/L (Ba, Cd, Ce, Cu, In, Mg, Pb, Rh, U). Through this method, the concentration ranges of each element were determined. For quantitative analysis, standard multi-elemental solutions of 10 μg/mL in 5% aqueous HNO_3_ (ICP-MS Calibration Standard 2 containing Ce, Dy, Er, Eu, Gd, Ho, La, Lu, Nd, Pr, Sm, Sc, Tb, Th, Tm, Y, and Yb) and ICP-MS Calibration Standard 3, containing Ag, Al, As, Ba, Be, Bi, Ca, Cd, Co, Cr, Cs, Cu, Fe, Ga, In, K, Li, Mg, Mn, Na, Ni, Pb, Rb, Se, Sr, Tl, U, V, and Zn, and 10 mg/L in 1% aqueous HNO_3_ (ICP-MS Calibration Standard 4 containing Au, Hf, Ir, Pd, Pt, Rh, Ru, Sb, Sn, and Te) were prepared by adding 2% ethanol to all stock standard solutions. These standards were prepared by taking into account the semi quantitative method mentioned above, for each concentration level of analyte. Thus, all standard solutions were matched with the distillate matrix (2% HNO_3_, 2% ethanol), all (*v*/*v*), taking into account the sample dilution degree and matrix interferences. A mixture of 2% HNO_3_ and 2% ethanol were used as blank solutions. All samples were measured in duplicate.

### 2.4. Statistical Data Processing

Statistical data preparation was made using SPSS Statistics 24 software (IBM, New York, USA). All measured experimental parameters were used as a single matrix for distillate classification regarding the fruit varieties—the geographical and specific trademark fingerprints. Principal component analysis (PCA) is one of the most widely employed unsupervised methods that aims to transform the original variables into new, uncorrelated variables [12]. The new principal components (PCs) obtained are linear combinations of the original variables. PCA provides information of the most meaningful parameters, in this specific case, the elemental content of fruit distillates, which describe the whole dataset interpretation, with minimum loss of the original information [13]. To fulfill the aforementioned classifications, linear discriminant analysis (LDA) was applied. This method is a supervised technique, which means that samples are coded from the beginning of the analysis, according to the label. Furthermore, the method attempts to find the optimal parameters that are able to maximize the distances between predefined groups, and to minimize the distances within the same group. The predictors are comprised in a linear function, called discriminant functions (DF). Based on these functions, all samples are classified, and the result is expressed as percentages. Moreover, a cross-validation is made, which implies testing each sample using a model obtained from the rest of the samples. A higher value obtained from the cross-validation procedure suggests the robustness of the model.

## 3. Results

Figure 1 presents the variation limits and the mean values of the isotope ratios (δ^18^O vs. (D/H)_I_), determined on distinct fruit distillates (plums, pears, apples, grapes, apricots, quinces, and bitter cherries). As can be seen, a clear separation among the investigated fruit distillates, with respect to the fruit variety, was not achieved. Among the investigated distillates, the higher mean δ^18^O value from ethanol was obtained for grapes and apricots. This result is explained by differences in the δ^18^O values of vegetal water, which characterized each fruit, and is in good accordance with the results reported by other authors [10]. These results are also in agreement with our previously reported studies on the isotope fingerprint of Transylvanian fruit juices [14], in which more elevated δ^18^O values of fruit waters extracted from grapes, as compared with those from apples and pears, were found. On the other hand, the determined values of the isotopic ratios (D/H), obtained through NMR measurements of deuterium in natural abundance on the ethanol extracted from fruit spirits, were between 96.83 and 103.89 and 121.94 and 129.83 ppm for (D/H)_I_ and (D/H)_II_, respectively (Table A1). Thus, the average values of our investigated distillates were comparable to those reported in the literature [15], where the values obtained for distinct types of fruits spirits were pears–100.21/125.1; apples–97.47/123.07; apricots–100.33/126.78; plums–8.33/125.63, values in ppm for (D/H)_I_ and (D/H)_II_, respectively.

Plants can be divided into two categories, depending on the photosynthetic pathway. Most plants (potato, rice, wheat, majority of grains and fruits) follow the C_3_ photosynthetic pathway, having carbon stable isotope composition, δ^13^C, between −34‰ and −22‰ [16]. Maize, sugarcane, sorghum, millet follow the C_4_ pathway, having δ^13^C values between −16‰ and −9‰. As a result, different ^13^C isotopic fingerprints are found in products from C_3_ or C_4_ plants, with the δ^13^C value being used to distinguish the carbon botanical origin. For the investigated fruits spirits, δ^13^C values ranged between −27.9‰ and −18.6‰. Thus, based on the δ^13^C value, it can be stated that three of the samples (pear, black currant, and bitter cherry) were identified as containing sugar addition from C_4_ plants (−21.9‰; −21.1 ‰, and −18.6‰, respectively). These results were also confirmed by NMR analysis, but also by the fruit distillates processors. The sugar addition could be explained through the fact that the raw materials used to produce fruit spirits generally must contain a high concentration of natural sugars, as is the case of plums. However, there are specific cases when the fruits chosen for their special taste and flavor (e.g., blackcurrant or bitter cherry), do not have sufficient natural sugar available for enzyme action, and the sugar concentration is adjusted by its supplementary addition in order to produce the fermentation.

Apart from the subtle differences, which were recorded in terms of isotope fingerprint among distinct fruit distillate types, an important differentiation factor is represented by the elemental content (Table A1). It is believed that this elemental profile is mainly influenced by the elaboration process of the fruit distillates, especially for elements such as Cu and Pb, mostly derived from the distillation equipment [17]. Therefore, based on the elemental profile of the investigated distillates, there were attempts to trace the influences of producer technologies (distillation and storage), based on the investigated parameters. Taking into account that most of the authentic samples were provided by two processors and one traditional producer (manufacture), for this classification, only these three distillate providers were considered. By applying PCA (Figure 2), we noticed that the first two principal components contributed to 48.23% of the variance (PC 1–33.83%, PC 2–14.4%). Furthermore, we observed that using the association between the metal content and unsupervised statistical methods, a perfect separation could not be obtained. Apart from this, we noticed that two distillate samples were totally separated from the main cluster, suggesting that these spirits were different from the rest. By verifying the provenience of these two samples, we found that these two were commercialized by Producer 1, and were special samples: one of them was a beer distillate and the other was an aged grape distillate (30 years old). The beer was produced following an artisanal process in a distinct manufacture and only the beer distillation was made by Producer 1. Therefore, the influences of the beer signature were observed in the final distillate. Concerning the aged grape distillate, this was only aged by Producer 1; the raw materials and the distillation were performed in the South part of the country.

Another important representation resulting from applying PCA is the loading plot (Figure 3), where each variable receives a coefficient correlated to its impact upon the sample set. Thus, variables with higher influence have higher coefficients and are situated far from the origin; the parameters with lower coefficients have slighter effects, and are much closer to the origin. All variables are grouped within principal components (PCs), the first PCs being the most important, with the highest eigenvalues. Usually, only the first two or three components (which have eigenvalues higher than 1) are used for further interpretations. In this case, PC1 has an eigenvalue of 6.67, while the second has a lower eigenvalue of 2.88, but still with a statistical contribution. The first PC has high loadings for Li (0.919), Mn (0.905), and Rb (0.957), and could represent the geological influence upon sample distribution. This is in good accordance with our previously published works [18], in which Li and Mn were found to be among the most representative markers for the geographical origin. The same PC has moderate loadings for Al (0.886) and for Cd (0.850), while the second PC has only one high value for As (0.940) and a moderate value for Se (0.864). After a visual inspection of both representations, it can be stated that the two samples that are far from their groups have high content of As and Se, and Li, Mn, and Rb, respectively. As previously stated, the geographical origin of the raw materials that were used for the production of these spirit samples are distinct from the rest, and this fact is very nicely pointed out through some of the most powerful markers in this regard, such as Li, Mn, and Rb. The efficiency of Li and Mn for geographical origin assessment was also reported for other matrices, such as coffee beans [19]. Apart from this, the As content was previously found as being a geographical differentiation marker for other commodities (i.e., potatoes) grown in distinct areas of Transylvania [20].

A good separation among distillates, with respect to their botanical and geographical origins or trademarks, could not be obtained based on the direct correlation of some determined isotope values (Figure 1) or by using unsupervised statistical methods (Figure 2 and Figure 3). Therefore, to achieve reliable discrimination among the fruit distillates, with respect to distinct criteria and based on all determined parameters, a supervised statistical treatment on all experimental data was further applied.

## 4. Discussion

Compared to other matrices, such as vegetables, fruits, and honey [18,20,21,22], the differentiation of fruit distillates, with respect to various criteria, such as geographical or botanical origins, based on isotope and elemental content, is not straightforward. This is mainly due to the extensively technological processes in which these products undergo, affecting the original fingerprint of the raw materials (i.e., fruits), and inducing, especially in the case of the elemental profile, a producer signature. Therefore, the aim of this pilot study was to identify the best isotope and elemental markers association that could discriminate among distinct fruit varieties, geographical origins, and trademarks of fruit distillates, using as a statistical tool a supervised chemometric method, namely linear discriminant analysis (LDA).

### 4.1. Geographical Differentiation

The first performed differentiation was realized, having as discrimination criteria the geographical origin of the fruits that were used as raw materials for the elaboration of distillate samples, in order to identify a more specific regional fingerprint inside the Transylvania region. To achieve this, and because of the sample distribution per area, the distillates set was split into three groups, corresponding to two geographic regions, namely Satu Mare and Bistriţa, and another group containing samples from other Transylvanian areas. By applying LDA, the obtained separation in the initial classification was 96.2%, while for the cross-validation procedure, a percentage of 84.6% was achieved (Figure 4). Because three classes were compared, the discrimination was made based on two discriminant functions: (DF1) and (DF2), respectively. The first function (DF1) explains the classification in a percentage of 72.8% and comprises the following predictors: K, Cu, and δ^18^O. The second function (DF2) explains the remaining 27.2%, and has a main discriminator B. The overlaps appear among Satu Mare and Bistriţa groups; three samples were misclassified, namely, two samples from Bistriţa were placed upon the Satu Mare group and one sample belonging to Bistriţa was attributed to Satu Mare.

Among the obtained differentiation markers, the presence of δ^18^O from ethanol (an acknowledged marker for the geographical origin) was noticed [10]. This is because oxygen atoms from ethanol are also derived from the fermentation water [23], which is directly related to the geographical origin of the raw materials (i.e., fruits). K and Cu were also found as differentiation markers for the geographical origin of honey [22] and might be related to agricultural practice. In their review paper, Bai, Shen, and Huang [24] explained that the use of potassium (K) fertilizer for fruits (pear, apple, cherry, etc.) leads to better development of the plants, increases nutritional concentrations of fruit, and prevents the appearance of certain destructive diseases.

Boron and Cu, together with Fe, Mn, Zn, Mo, Ni, and Cl, are part of the micronutrient group required for the plant’s growth [25]. Boron is directly involved in plant metabolism and has an important function in pollen germination and pollen tube growth [24]. The effect of boron fertilization on fruit trees, on fruit quality, has been well known for many years [25,26]. “Fruiting” always requires boron, especially in the case of small stone plants, because this small stone fails to provide the necessary amount for the plant. It was reported in the literature [27] that boron and copper were also found as differentiation markers for wine samples, among other elements, such as Cs, Mn, and Rb, for an extended area (around 1000 km^2^) from South Africa. Apart from this, boron was observed among differentiation predictors of rice samples cultivated in the USA, Europe, and Basmati region [28].

From the above presented results, the suitability of this approach for the differentiation of fruit distillate origins could be seen, even for areas located inside of a region, in our case Transylvania. This can also have commercial importance in consumer perceptions, which associate specific tests and quality with certain areas.

### 4.2. Trademark Specific Fingerprint

Based on the multiple production steps that are conducted during the fruit spirit elaboration, including the preparation technology and aging in different wood barrels, a trademark fingerprint is expected to exist. Moreover, this trademark fingerprint was previously pointed out in our previous study [6], where the association between Raman spectroscopy and machine learning algorithms allowed us to prove the existence of a specific producer signature. Therefore, starting from this, we intended to verify if such a fingerprint could also be pointed out through the association between the isotope values and elemental composition in association with supervised statistical methods. For this aim, samples from two distillates processors and one manufacture were analyzed and statistically processed using LDA. Based on this approach, it was possible to identify a trademark fingerprint, mainly given by the above-mentioned influences.

Thus, the simultaneous differentiation among the three fruit distillate producers was realized in a percentage of 100% in the initial classification, while in the cross-validation procedure, a percentage of 95.7% was reached (Figure 5). One single sample from Producer 2 was misclassified in this case, being attributed to Producer 3 (manufacture). This differentiation was made based on two prediction functions, which explained the classification in a percentage of 53.7% (DF1) and 46.3% (DF2), respectively. The main predictors from DF1 were Ni (−1.216), Mo (1.230), Cu (1.152), B (−0.891), and Tb (0.974) from DF2: K (−1.399). Some of these predictors, such as Ni and Cu, are related to the specific conditions, which are followed by each producer during the distillation process and are directly influenced by the equipment involved in the technological processes. On the other hand, K and B content might be influenced by the growing regions of the used fruits [18], regions that are also related to the distillate producers. It can be observed that, among these markers, those found for geographical differentiation are included, namely the content of K, Cu, and B. This is because most of the raw materials, which were used by two of the distillate producers, came from their specific areas: Producer 1—Satu Mare; Producer 2—Bistriţa. In the case of the distillate samples purchased from the manufacture, the raw materials originated from different areas in Transylvania. Therefore, in this case, overlapped influences were given by the distinct geographical origin, as well as different distillate production processes conducted to the present separation. These results have a high practical applicability in studies related to brand protection. This is mainly because these commodities have a high market value and represent a specific product of Transylvania. 

### 4.3. Fruit Variety Differentiation of Raw Materials Used for Fruit Distillates

Because of the limited sample numbers per class, the individual distillate fingerprints, with regard to fruit variety, were identified by performing the classifications between two groups, i.e., the investigated fruit spirit versus all of the rest. Only the distillate varieties for which we had at least three samples per class were discussed. Based on this approach, it was possible to assess the main predictors, which differentiated a certain distillate variety type from the rest.

The best-obtained differentiation was achieved for apricot and pear distillates for which a correct classification percentage of 100% was obtained in both classifications (initial and cross-validation). For apricot spirits, the differentiation was obtained based on the following markers: Co (2.378), As (3.518), Gd (2.065), Tm (−1.045), while for pear distillates, the predictors were Ba (1.795), Er (−2.934), and Yb (2.839). Most likely, these elements appeared from the agricultural chemicals, which were used during the fruit growth processes [29].

The plum distillate differentiation was realized at percentages of 88.2% and 82.4% in the initial and cross-validation procedures based on Li and Ni content as discrimination markers. Ni content was also found to be a discriminator for honey botanical origin [22,30], despite it not being a specific marker for the botanical origin of honey, being mainly related to pollution issues and pesticides. A similar percentage in the cross-validation procedure (82.4%) was achieved for apple spirits, and the differentiation parameter was, in this case, the ratio (D/H)_II_. This ratio is mainly related to the meteorological conditions and, to a lesser extent, the sugar content of the apples. The weaker obtained discrimination was the one obtained for quince spirits, around 70% (76.5% in initial classification and 70.4% in the cross-validation procedure), having as a discrimination marker the calcium content. Indeed, for quince spirits, the mean Ca value was higher as compared to other distillates (i.e., apple distillates). This fact can also be explained through the high calcium content of quince fruits as compared to others, such as apples [31]. Taking into account that this study is a preliminary one, another study, in which a higher number of samples per class, needs to be performed for varietal discrimination. 

## 5. Conclusions

This work confirms the potential association between isotope and elemental content in conjunction with supervised statistical methods, namely linear discriminant analysis (LDA), for spirits differentiation. The effectiveness of this approach was mainly proved for the trademark fingerprint differentiation, which was achieved at a percentage of 100% for the initial classification and 95.7% in the cross-validation procedure, respectively. This classification was mainly based on the following elements: Ni, Mo, Cu, B, Tb, and K. Three of these discriminators (K, Cu, and B) were among the most powerful markers for the geographical origin differentiation. The explanation for this overlapping is due to the distillate producers utilized for the spirit elaboration, mainly raw materials from specific areas of Transylvania; thus, the geographical fingerprint was included in the trademark one. Apart from these elements, as expected, the isotope value δ^18^O also proved to be a good geographical indicator.

Despite the fruit spirit matrix being over-processed, it was possible to point out the fruit fingerprint via this approach. Thus, the individual discrimination of apricot and pear distillates from the other spirits was achieved at a percentage of 100% in both initial and cross-validation procedures. On the opposite side was the quince distillate differentiation, where a modest percentage was obtained (76.5% in the initial classification followed by 70.4% in cross-validation).

Based on the promising results obtained in this pilot study, further research that involves a higher number of samples is needed.

## Figures and Tables

**Figure 1 foods-10-03000-f001:**
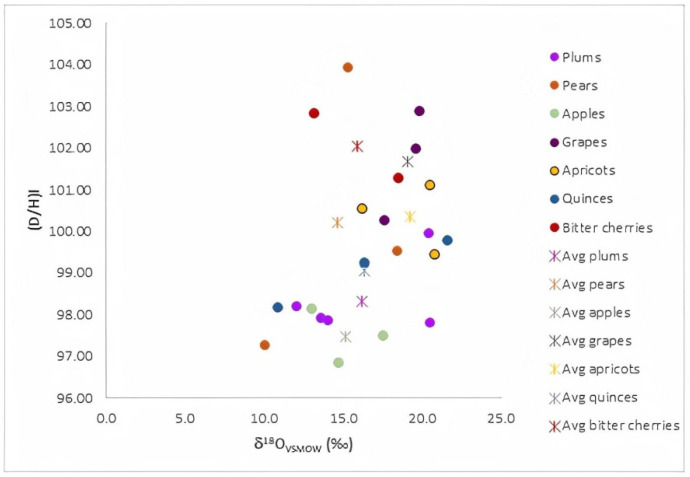
Distribution of δ^18^O and (D/H)_I_ value characteristics to distinct fruit distillates (plums, pears, apples, grapes, apricots, quinces, and bitter cherries), along with their corresponding varietal mean values.

**Figure 2 foods-10-03000-f002:**
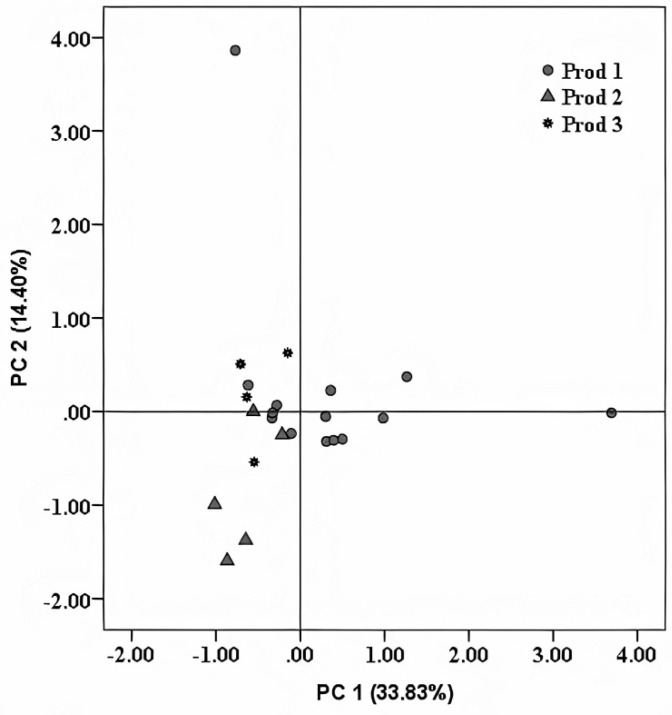
Score plot obtained after applying PCA on the elemental content of investigated fruit distillates.

**Figure 3 foods-10-03000-f003:**
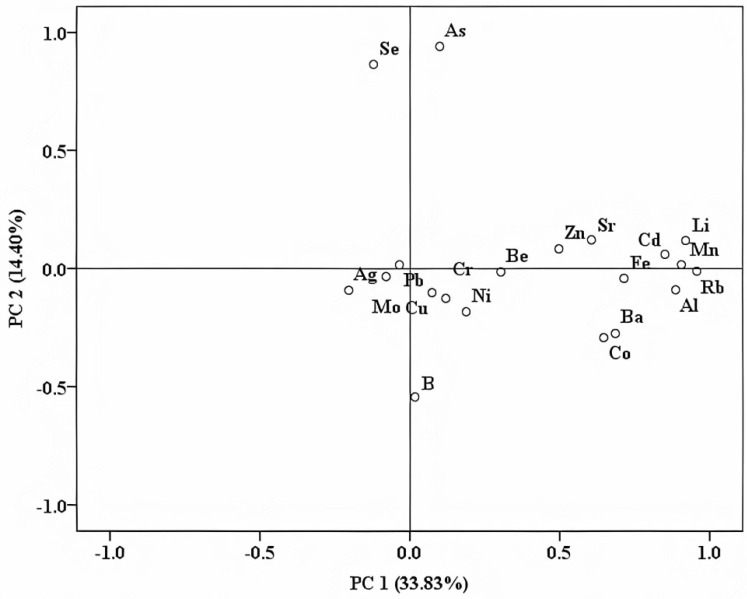
Loading plot obtained after applying PCA on the elemental content of all fruit distillate samples.

**Figure 4 foods-10-03000-f004:**
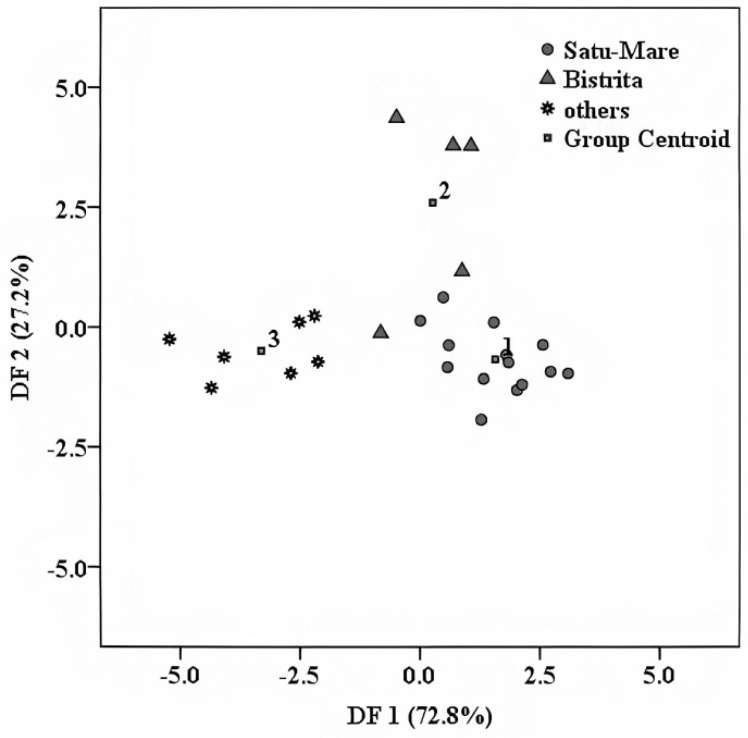
Geographical differentiation of the investigated fruit spirits based on LDA.

**Figure 5 foods-10-03000-f005:**
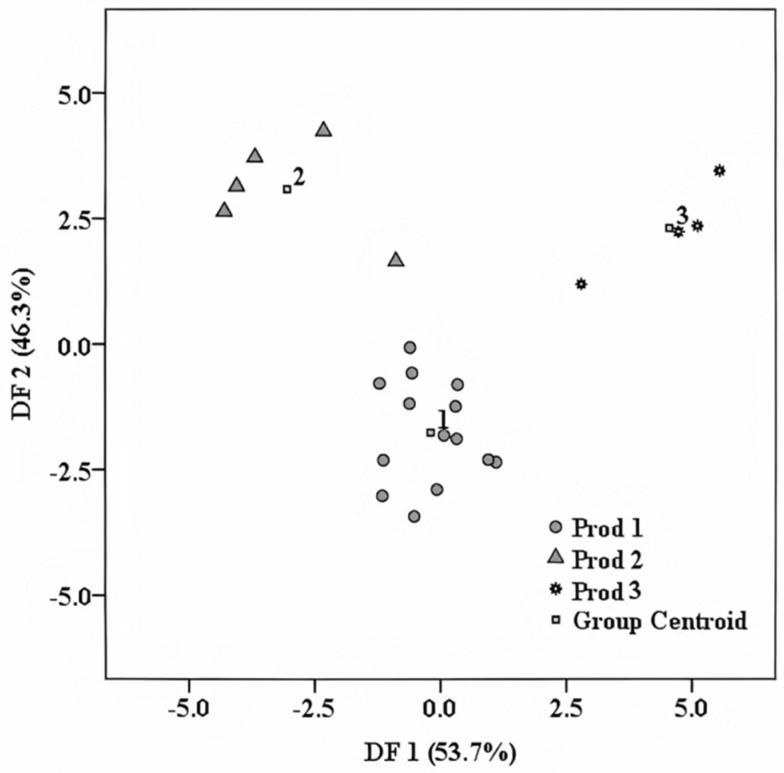
Trademark identification of the investigated distillates based on LDA.

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
