# Peer review of "Authentication of Transylvanian Spirits Based on Isotope and Elemental Signatures in Conjunction with Statistical Methods"

_foods, 2021, doi:10.3390/foods10123000_

Round 1

Reviewer 1 Report

2.1. sample description. Perhaps mentioning latin names of the species would eb benefficial. In case of quince, blueberries (or bilberries??) would be useful in understand what was the actual type of the specific fruit used. however, it could be hard determining this if those are commercial samples.

The quality of the figures should be improved (resolution).

Was the data normalised or standartised before the PCA analysis?

I would suggest using the same colours for each of the beverages in Figures throughout the manuscript.

PCA loadings and scroe plots could be easily combined - would be easier to see the relations between the investigated parameters.

Figure 5 - producer 3 is with the small letter

The main concern with this article is that the title mentions use of isotope ratios for authenticity testing, however, in reality there is very little about the use of IRMS, but more on the elemental analysis. Please provide more on the isotope analysis. Show also the analysis of C which could possibly show differences between grain and berry/fruit alcohols. I also dont see much on the results from the NMR analysis? 

Overall the article is interesting- the used statistics are appropriate, hwoever, the methods described are not reflected in the results, which is crucial for the article to be considered as finished and whole.

Author Response

Thank you for all your effort related to the improvement of the submitted manuscript. We are presenting below our answers to all observations:

  1. 1. sample description. Perhaps mentioning latin names of the species would eb benefficial. In case of quince, blueberries (or bilberries??) would be useful in understand what was the actual type of the specific fruit used. however, it could be hard determining this if those are commercial samples.

Answer

            The Latin names of species were provided in the manuscript (please see the text).

  1. The quality of the figures should be improved (resolution).

Answer

      The quality of all figures has been improved and the enhanced figures have been introduced in the manuscript.

  1. Was the data normalised or standartised before the PCA analysis?

Answer

Yes, the experimental data matrix was normalized before running PCA.

  1. I would suggest using the same colours for each of the beverages in Figures throughout the manuscript.

Answer

The utilized software for the data processing allows a single label to be illustrated for a point (sample) on a plot, at a time (i.e. the label corresponding to the fruit type, trademark, or geographical origin). Thus, by applying the supervised method (linear discriminant analysis), the aim was to create models that are able to differentiate the samples according to a specific classification type and, therefore, the labels were chosen accordingly. For example, the points plotted with respect to the discriminant functions, obtained when the trademark differentiation was desired, were characterized by a shape that referred to one of the classes: Prod 1, Prod 2, Prod 3.

  1. PCA loadings and scroe plots could be easily combined - would be easier to see the relations between the investigated parameters.

Answer

The SPSS software used for statistical data interpretation does not have the option for constructing a biplot representation.

  1. Figure 5 - producer 3 is with the small letter

Answer

The small letter was corrected.

  1. The main concern with this article is that the title mentions use of isotope ratios for authenticity testing, however, in reality there is very little about the use of IRMS, but more on the elemental analysis. Please provide more on the isotope analysis. Show also the analysis of C which could possibly show differences between grain and berry/fruit alcohols.

Answer

We agree with the reviewer observation and we added a paragraph related to the application of δ13C value in fruit distillates differentiation (please see the manuscript).

  1. I also dont see much on the results from the NMR analysis? 

Answer

As was suggested, a comparison of our results with those previously reported in the literature was added in the manuscript.

Reviewer 2 Report

This paper examines the origins of spirits from the Transylvanian region. Stable isotopes, elemental ratios, magnetic resonance imaging and statistical analysis are used to identify the origin. The work is really applied, it allows to better characterize the origin of spirits.

Nevertheless, in this work, I lacked clarity on what the authors want to demonstrate. Is it to say that spirits from this region are unique and different (e.g. in terms of soil elemental composition, water isotope ratio) from other regions, or that factories that produce spirits are unique in their pots and tubes, which give a unique microelement ratio?

Can we say that the three producers (who are studied in this work) are characterized by the products they produce now? And if they replace the pipes due to technological processes,  the elemental composition probably will change? I would like the authors to consider such a scenario in the discussion part.

I suggest the authors change the manuscript, highlighting the places where the origin is determined by 1)  the geographical signal, 2) by the industrial signal, and finally 3) when everything is combined. I understand that using geographic signal alone can be a weak separation, but it is precisely the importance of combining the article and the methods demonstrated.

Author Response

Thank you for all your effort related to the improvement of the submitted manuscript. We are presenting below our answers to all observations:

  1. Nevertheless, in this work, I lacked clarity on what the authors want to demonstrate. Is it to say that spirits from this region are unique and different (e.g. in terms of soil elemental composition, water isotope ratio) from other regions, or that factories that produce spirits are unique in their pots and tubes, which give a unique microelement ratio?

Answer

            The main aim of our study was to demonstrate that the approach consisting in the association between the isotopic and elemental profile of distillates, and advanced chemometric methods can discriminate the fruit distillates according to distinct criteria. Thus, it was possible to differentiate the fruit spirits according to their geographical and botanical origin. Moreover, it was possible to point out a trademark fingerprint that was reported in this study, to our knowledge, for the first time in the literature. In terms of geographical differentiation, it is important to emphasize here that the fruit distillates differentiation was possible to be made inside of a quite restricted region (Transylvania) thus demonstrating the potential of this approach for this purpose.  

  1. Can we say that the three producers (who are studied in this work) are characterized by the products they produce now? And if they replace the pipes due to technological processes,  the elemental composition probably will change? I would like the authors to consider such a scenario in the discussion part.

Answer

In the case of distillates processors, which we included in this study, their aim is to preserve the traditional way of distillates obtainment, therefore they are not changing too many steps/equipment from one year to another. We agree with the reviewer's observation and we rewrite the phrase from rows 350-351 in order to sustain this idea.

  1. I suggest the authors change the manuscript, highlighting the places where the origin is determined by 1)  the geographical signal, 2) by the industrial signal, and finally 3) when everything is combined. I understand that using geographic signal alone can be a weak separation, but it is precisely the importance of combining the article and the methods demonstrated.

Answer

Due to the fact that most of the raw materials that were used by each producer were mainly local, a complete separation of the geographical and technological processes cannot be made. This is the reason for which some commune discrimination markers for geographical and trademark fingerprint were obtained. This suggestion is nevertheless a very good one and it will be taken into account for our future work in which a most appropriate sample distribution for this purpose will be used.